# Annexins in Adipose Tissue: Novel Players in Obesity

**DOI:** 10.3390/ijms20143449

**Published:** 2019-07-13

**Authors:** Thomas Grewal, Carlos Enrich, Carles Rentero, Christa Buechler

**Affiliations:** 1School of Pharmacy, Faculty of Medicine and Health, University of Sydney, Sydney, NSW 2006, Australia; 2Department of Biomedicine, Unit of Cell Biology, Faculty of Medicine and Health Sciences, University of Barcelona, 08036 Barcelona, Spain; 3Centre de Recerca Biomèdica CELLEX, Institut d’Investigacions Biomèdiques August Pi i Sunyer (IDIBAPS), 08036 Barcelona, Spain; 4Department of Internal Medicine I, Regensburg University Hospital, 93053 Regensburg, Germany

**Keywords:** annexins, adipose tissue, adiponectin, cholesterol, glucose homeostasis, inflammation, insulin, lipid metabolism, obesity, triglycerides

## Abstract

Obesity and the associated comorbidities are a growing health threat worldwide. Adipose tissue dysfunction, impaired adipokine activity, and inflammation are central to metabolic diseases related to obesity. In particular, the excess storage of lipids in adipose tissues disturbs cellular homeostasis. Amongst others, organelle function and cell signaling, often related to the altered composition of specialized membrane microdomains (lipid rafts), are affected. Within this context, the conserved family of annexins are well known to associate with membranes in a calcium (Ca^2+^)- and phospholipid-dependent manner in order to regulate membrane-related events, such as trafficking in endo- and exocytosis and membrane microdomain organization. These multiple activities of annexins are facilitated through their diverse interactions with a plethora of lipids and proteins, often in different cellular locations and with consequences for the activity of receptors, transporters, metabolic enzymes, and signaling complexes. While increasing evidence points at the function of annexins in lipid homeostasis and cell metabolism in various cells and organs, their role in adipose tissue, obesity and related metabolic diseases is still not well understood. Annexin A1 (AnxA1) is a potent pro-resolving mediator affecting the regulation of body weight and metabolic health. Relevant for glucose metabolism and fatty acid uptake in adipose tissue, several studies suggest AnxA2 to contribute to coordinate glucose transporter type 4 (GLUT4) translocation and to associate with the fatty acid transporter CD36. On the other hand, AnxA6 has been linked to the control of adipocyte lipolysis and adiponectin release. In addition, several other annexins are expressed in fat tissues, yet their roles in adipocytes are less well examined. The current review article summarizes studies on the expression of annexins in adipocytes and in obesity. Research efforts investigating the potential role of annexins in fat tissue relevant to health and metabolic disease are discussed.

## 1. Introduction

### 1.1. Obesity

In most countries, the increasing prevalence of obesity represents a rapidly growing risk factor for chronic liver diseases, type 2 diabetes (T2D), cardiovascular diseases and most types of cancer. The mechanisms contributing to obesity are multifactorial and are far from being completely understood. Moreover, life style changes with less caloric intake and increased energy expenditure appear insufficient to reduce body weight in the long term. Hence, the identification of the multiple processes that contribute to excess adiposity is required to enact innovative strategies to combat this epidemic [1,2]. Some key features and cellular machineries that contribute to increased and dysfunctional fat mass are listed below.

Adipose tissue is central to the development of obesity and is composed of different cell populations including fibroblasts, preadipocytes, mature adipocytes, macrophages, mesenchymal stem cells, endothelial cells, and vascular smooth muscle cells, with cellular function as well as their quantity being affected by obesity [1]. All these different cell types, through various mechanisms, contribute to obesity and associated comorbidities, which has been reviewed in detail elsewhere [1,2,3]. In brief, storage of nutrients and their mobilization for energy production are critical functions of adipose tissue. Yet, increased lipolysis in obese fat tissue is closely associated with the development of insulin resistance and T2D [1,2,3]. In addition, the excessive accumulation of fat in adipocytes due to overnutrition can lead to an inflammatory response that creates further metabolic complications [3,4,5]. In fact, even the physical stress triggered by the swelling that occurs in adipocytes upon increased fat accumulation seems to contribute to inflammation and insulin resistance [6]. In regard to the inflammatory process, macrophages accumulate in adiposity and in response to environmental signals in the fat tissue, undergo polarization to pro-inflammatory M1 macrophages [3]. In addition, in adipose tissue other myeloid cells, as well as T- and B-lymphocytes, have been linked to macrophage homeostasis and the inflammatory process associated with obesity [7]. Moreover, the growing tissue is not appropriately supplied with oxygen causing hypoxia, which contributes to inflammation and fibrosis. This pathological progress, adipose tissue fibrosis, hinders tissue growth and is linked to metabolic impediments [3]. Further complexity is created by truncal or android fat distribution, which was recently identified as an independent risk factor for metabolic diseases in obesity. Also, visceral and subcutaneous adipose tissues differ in blood flow, cellular composition, adipocyte size and endocrine function, thereby contributing differently to whole body physiology [2,3]. 

Additionally, the identification of brown fat in humans [8,9,10] has initiated new exciting research in the field over the last decade, as its highly elevated expression of uncoupling proteins leads to the production of heat, which favors weight loss [3,11]. Due to its therapeutic potential, the process of browning has created great interest, where white fat cells become so-called beige or brite adipocytes, acquiring characteristics of brown fat, in particular the upregulation of uncoupling proteins. Hence, molecules targeting brown or brite fat to increase energy expenditure are being investigated for their potential to reduce body weight and improve metabolic health [12]. Actually, besides increased thermodynamic expenditure, the activation of brown adipose tissue additionally accelerated other cardioprotective and clinically relevant events, such as clearance of plasma triglycerides, a process that was dependent on the fatty acid transporter CD36 [13]. Furthermore, brown fat also contributed to lipoprotein processing and the conversion of cholesterol to bile acids in the liver, enabling the removal of excess cholesterol from the body [14]. Moreover, especially under thermogenic stimulation, brown fat releases several bioactive factors with endocrine properties, including insulin-like growth factor I, interleukin-6 (IL-6), or fibroblast growth factor-21, which influence hepatic and cardiac function, contributing to improved glucose tolerance and insulin sensitivity [15,16,17].

Given that obesity is characterized by an increased accumulation of triglycerides, research in the field over the last two decades has focussed on the dysregulation of the fatty acid metabolism. However, obese adipocytes also accumulate calcium and cholesterol crystals, which was demonstrated to contribute to oxidative stress and cell death [5]. On the other hand, plasma membrane cholesterol was depleted in obese fat cells which probably impaired the function of cholesterol-rich membrane microdomains (lipid rafts), causing an elevated release of C-C motif chemokine ligand 2 (CCL2), a major chemoattractant for monocytes [18]. In other studies, inhibition of the Niemann-Pick type C1 (NPC1) transporter, which facilitates cholesterol export from late endocytic (pre-lysosomal) and lysosomal compartments, impaired insulin signaling and glucose uptake in adipocytes [19]. Cholesterol is also essential for the proper functioning of endo- and exocytic vesicle transport, which control the release of distinct adipokines like adiponectin [20], an anti-inflammatory plasma protein that improves insulin sensitivity, but is reduced in obesity [21].

A more detailed analysis of the various pathways listed above and affected in obese adipose tissues clearly is essential to develop strategies to combat obesity. However, it would go beyond the scope of this review to list all pathways contributing to adipose tissue dysfunction and we refer the reader to other excellent articles [3,4,22]. In the following, we will summarize and focus on the current understanding of how a group of evolutionary conserved proteins, the annexins, may influence fat tissue function in health and disease.

### 1.2. Annexins

The annexin family in humans and vertebrates consists of twelve structurally related Ca^2+^- and membrane binding proteins (AnxA1–AnxA11, AnxA13) [23,24]. All annexins contain a variable N-terminus, followed by a conserved C-terminal domain with four (or eight in AnxA6) annexin repeats (Table 1). Each of these repeats encodes for Ca^2+^ binding sites, allowing annexins to rapidly translocate to phospholipid-containing membranes in response to Ca^2+^ elevation [23,25,26]. Hence, annexin functions are intimately dependent on their dynamic and reversible membrane binding behaviour. Nevertheless, their similar structure, phospholipid-binding properties, overlapping localizations, and shared interaction partners have made it difficult to elucidate their precise functions. Yet, despite in vivo studies in knock-out (KO) models strongly suggesting redundancy within the annexin family, specific functions of individual annexins have been identified [23,25,26,27,28,29,30,31]. Interestingly, besides often subtle differences in their spatio-temporal and Ca^2+^-sensitive membrane binding behaviour to negatively charged phospholipids, the diversity of N-terminal interaction partners, affinity to other lipids, including phosphatidylinositol-4,5-bisphosphate, cholesterol and ceramide, posttranslational modifications, and most relevant for this review, their differential expression patterns seem to facilitate opportunities to create functional diversity within the annexin family [23,25,26,27,28,29,30,31]. The subsequent chapters will review recent knowledge on the expression of individual annexins in adipose tissue, with quite diverse implications for adipocyte and macrophage function in health and obesity.

## 2. Annexin Expression Patterns in Adipose Tissue and Their Potential Functions in Obesity

### 2.1. Annexin A1 (AnxA1)

AnxA1 (previously known as lipocortin 1) is expressed in most cell types, and abundant in macrophages, neutrophils, the nervous and endocrine system [23,27,92]. Like other annexins, AnxA1 is found at multiple locations inside cells, including the plasma membrane, endosomal and secretory vesicles, the cytoskeleton and the nucleus, participating in membrane transport, signal transduction, actin dynamics and regulation of metabolic enzymes related to cell growth, differentiation, motility and apoptosis [25,26,92,93,94]. In addition, AnxA1 has a prominent extracellular function, acting as an anti-inflammatory, pro-resolving protein which exerts its effects via binding to the formyl peptide receptor 2 (FPR2). Both molecules are induced by glucocorticoids and contribute to the beneficial activities of these anti-inflammatory drugs [39,42].

The inflammation-related functions of FPR2 are diverse and complex, with multiple FPR2 ligands exercising various and sometimes opposite activities [36,95]. While the loss of FPR2 reduced inflammation, the overall FPR2 activity in fat tissue in vivo is most likely the net result of the distinct expression patterns and the localized distribution of different FPR2 ligands in this tissue [36]. Importantly, resolvin D1 and lipoxin A_4_, both bioactive lipid mediators that have been identified in adipose tissue, are agonists of this G-protein coupled receptor [96,97]. These lipids have anti-inflammatory activities and highlight the requirement to fine-tune the balance of ligands with opposing activities, in order to activate the immune response and thereby accelerate the termination of inflammation [96].

Recent studies suggest that the AnxA1/FPR2 axis is highly relevant for obesity and related inflammation, as well as other complications, such as insulin resistance, T2D and atherosclerosis [36,37,41,42,45]. As levels of FPR2 and its ligands critically influence strength of biological response, it is interesting to note that in obese mice, adipose tissue FPR2 mRNA and resolvin D1 levels were decreased [95]. Most relevant for AnxA1 in adipose tissue, the FPR2 peptide agonist WKYMVM, which is derived from the N-terminus of AnxA1, greatly enhanced the insulin response of diet-induced obese mice [45]. 

Somewhat unexpectedly, FPR2 deficiency improved the metabolic health of mice that were fed a high fat diet [36]. In this study, FPR2 was increased in fat of diet-induced obese mice and diabetic, leptin-receptor mutated, animals. Loss of FPR2 in macrophages blocked polarization into pro-inflammatory M1 macrophages [36]. FPR2 knock-out mice were less obese and higher thermogenesis in skeletal muscle was most likely responsible for enhanced energy expenditure [36]. Although the lack of FPR2 signalling events induced by ligands other than AnxA1 probably also contribute to the phenotype of the FPR2 knock-out mice described above, one can speculate that up- or downregulation of AnxA1 may also have profound effects on FPR2-dependent energy metabolism in adipose tissue.

In this context, it is still unclear which cell types contribute to extracellular AnxA1 levels in adipose tissue. In fat tissues, AnxA1 was more abundant in the stromal-vascular fraction than in adipocytes [43], indicating that infiltrating monocytes and macrophages expressing AnxA1 may represent the main source of extracellular AnxA1 in fat [39]. In support of this hypothesis, when these immune cells became activated, AnxA1 translocated to the cell surface and was secreted [39]. 

Besides the contribution of non-adipocytes to AnxA1 levels in fat mass, its expression appears tightly regulated during adipocyte differentiation, as murine 3T3-L1 adipogenesis identified AnxA1 mRNA and protein downregulation [44]. In contrast, in mature adipocytes from patients with Simpson Golabi Behmel syndrome, an overgrowth disorder leading to craniofacial, skeletal, cardiac, and renal abnormalities, AnxA1 mRNA and protein amounts were approximately 65-fold higher compared to their corresponding preadipocytes. As FPR2 levels were markedly reduced in this model, it remains to be determined if drastically upregulated AnxA1 expression alters the repertoire and availability of other extracellular FPR2 ligands and impacts on FPR2 activity [38,44]. Simpson Golabi Behmel syndrome is associated with glypican-3 loss-of-function mutations [98], implicating a possible link between adipocyte AnxA1 expression and this poorly characterized cell surface proteoglycan. However, a more likely explanation could be the higher concentration of glucocorticoids used in this study, possibly causing an elevation of AnxA1 levels irrespective of adipogenesis. The analysis of purified preadipocytes and mature cells may be an appropriate approach to better define transcriptional and post-transcriptional regulation of AnxA1 expression during adipogenesis. 

The therapeutic potential of AnxA1 is further underscored by its upregulation in the subcutaneous fat of obese men given rosiglitazone for two weeks [33]. Glitazones are insulin sensitizers and agonists of peroxisomal proliferator-activated receptor-γ (PPARγ), a master regulator of adipogenesis [99]. AnxA1 is a target gene of this transcription factor in breast cancer cells [100] and most likely in numerous other cell types [39,101]. Whether this PPARγ-dependent transcriptional control of the AnxA1 promoter also applies for adipocytes needs additional studies. 

Further documenting a relationship between AnxA1 and obesity, AnxA1 mRNA was strongly increased in adipose tissue of mice on a high fat diet [34]. This upregulation was observed in both leptin- and IL-6-deficient animals, strongly pointing at transcriptional pathways not directly regulated by these factors being responsible for AnxA1 upregulation in a lipid-rich environment [34]. AnxA1 mRNA expression was also higher in visceral adipose tissues of obese compared to lean children [102]. Proteome assessment of adipocytes isolated from subcutaneous fat of young and old overweight patients revealed higher AnxA1 protein levels in the latter [35]. Hence, as older subjects more often suffer from insulin resistance and cardiovascular disease, these findings further support a function of AnxA1 in metabolic health. Interestingly, under inflammatory conditions, AnxA1 may undergo protease-mediated degradation, leading to pro-inflammatory AnxA1 fragments that lack the FPR2-binding motif in the N-terminal AnxA1 region [103]. Indeed, cleaved AnxA1 was more abundant in adipose tissues of obese individuals independent of their insulin resistance status [40].

Whole body physiology critically influences adipose tissue function and in the following, we will briefly summarize some observations that could impact on AnxA1 levels and functions in fat tissue. In contrast to upregulated adipose AnxA1 levels in obesity-related disease settings listed above [33,34,102], one study identified that circulating levels of AnxA1 were decreased in obesity [38]. Yet, more recent research described that serum AnxA1 amounts increased with body mass index (BMI) and positively correlated with IL-6 [40]. In the same report, an association of serum AnxA1 levels with T2D was not apparent [40]. The opposing outcome of these two studies clearly illustrates that further research is needed to resolve the role of AnxA1 in adiposity and metabolic diseases. 

Over the years, many studies have established that dysregulation of the inter-organ cross-talk beween adipose tissue and other metabolic organs contribute to significant changes in energy homeostasis, glucose and lipid metabolism in obesity and associated complications. Adipose tissue releases numerous adipokines that influence liver, muscle and pancreas physiology, which in turn, have potential to modify glucose and lipid handling in fat tissue [3,4]. This may also include alterations in AnxA1 expression, secretion, and protein stability, which may impact on serum AnxA1 levels or influence other AnxA1-related biological activities with indirect effects on adipocytes. For example, non-alcoholic fatty liver disease is commonly diagnosed in the obese and is a spectrum ranging from benign liver steatosis to hepatitis and fibrosis [21]. Hepatic AnxA1 protein expression was reduced in patients with bridging fibrosis when compared to those with mild disease [104]. In mice fed a methionine-choline-deficient diet to induce non-alcoholic steatohepatitis (NASH), hepatic AnxA1 protein levels were nevertheless increased [104]. While these findings may suggest a link between AnxA1 expression levels and hepatic neutral lipid accumulation, oleate-induced lipid storage was normal in AnxA1-overexpressing Huh7 hepatocytes [61]. Accordingly, hepatic triglycerides levels were also comparably induced in murine NASH of wild type and AnxA1-deficient mice [104]. Yet irrespective of neutral lipid storage, liver inflammation and fibrosis were clearly enhanced in AnxA1 KO-animals [104]. 

AnxA1 was expressed in liver macrophages and contributed to anti-inflammatory M2 macrophage polarization and IL-10 production. Accordingly, macrophages developed into a pro-inflammatory M1 phenotype in the AnxA1 null animals [104]. Galectin-3 is produced by activated macrophages and contributes to liver fibrosis, and recombinant AnxA1 prevented galectin-3 expression [104]. Strikingly, AnxA1 protected the liver from NASH in this experimental model, which is characterized by body weight loss [104]. Furthermore, inhibition of hepatitis C virus replication by AnxA1 showed a protective role in the development of chronic liver disease [105]. Again, steatosis grade was not changed by AnxA1 in the liver cells [105]. Hence, these studies suggest protective roles for AnxA1 in liver function, which could also support a healthy communication with adipose tissue.

Beneficial effects of AnxA1 were also described in muscle and pancreatic beta-cells, both highly relevant for glucose homeostasis [106]. The saturated fatty acid palmitate, which is elevated in the plasma of obese patients, induced insulin resistance and suppressed AnxA1 expression in L6 myotubes [45,107]. On the other hand, AnxA1 released from mesenchymal stromal cells improved the glucose-induced insulin release of human islets in a co-culture model demonstrating protective functions on pancreatic beta-cells [108]. 

Taken together, most of the data summarized above point towards disease-preventing activities of AnxA1 in obesity (Table 1). In further support of this model, AnxA1 null mice were in fact more obese, had larger adipocytes and increased leptin levels when fed a high fat diet [34]. Common measures that occur with high fat diet feeding, such as upregulation of lipolytic enzymes and downregulation of 11-beta hydroxysteroid dehydrogenase type 1, was only significant in fat tissues of the obese wild type animals [34]. Corticosterone levels were higher in the AnxA1-deficient animals and may have further promoted adiposity in these mice [34]. Moreover, the high fat diet fed AnxA1 KO-mice displayed elevated glucose and insulin levels, and were less insulin-sensitive. Interestingly, despite the prominent anti-inflammatory features of AnxA1 discussed above, adipose tissue inflammation was not induced in these mice [34]. The exacerbation of obesity-associated metabolic diseases in AnxA1 null mice was confirmed in a further study. The treatment of these mice with recombinant human AnxA1 reduced body weight, fat mass, and liver steatosis [41]. 

Finally, others analyzed AnxA1 null mice fed a control chow diet. Body weight and adipocyte size were normal, whereas epididymal fat mass was reduced in AnxA1-deficient animals [43]. Catecholamine-induced rise in cAMP levels and lipolysis were more pronounced in adipose tissue explants of the control animals [43]. Adipose tissue explants from the AnxA1 KO-mice further displayed a lower production of IL-6, which was not attributed to a decline in the number of macrophages in intra-abdominal fat pads [43]. 

Overall, the studies summarized above indicate AnxA1 as a metabolism-improving molecule in models of metabolically stressed animals (Figure 1, Table 1). This may provide exciting therapeutic opportunities [37,39,41,42,45,104], but more research—exploring for instance the comparison of energy expenditure measurements in controls and the AnxA1 null mice on chow and high fat diets [34,41]—is still needed to better understand the various molecular pathways regulated by AnxA1 in adipose tissues.

### 2.2. Annexin A2 (AnxA2)

AnxA2 is ubiquitously expressed and most abundant in endothelial cells, monocytes and macrophages. In addition, AnxA2 is also often upregulated in cancers [23,25,94,109,110,111]. Most AnxA2 proteins form a heterotetrameric complex with p11, a member of the S100 protein family, at the plasma membrane and intracellular compartments, while only small amounts of AnxA2 monomer are present in the cytosol, endosomes and nucleus. In these multiple locations, AnxA2 contributes to the regulation of endo-/exocytic membrane transport, microdomain organization, membrane repair and nuclear transport, relevant for many different cellular activities [23,25,26,93,94,109]. Also, extracellular AnxA2 activities related to fibrinolysis and not discussed further in this review have been well documented [109,110,111,112].

AnxA2 is expressed in the adipose tissues of humans and rodents [53,57] and has been linked with two prominent aspects of adipocyte function (Table 1). Firstly, several studies implicated AnxA2 in glucose homeostasis, in particular the insulin-inducible translocation of GLUT4, the main glucose transporter in adipocytes, from intracellular compartments to the cell surface. In one study, the silencing of AnxA2 in 3T3-L1 adipocytes improved insulin sensitivity and glucose uptake [59]. In striking contrast, others reported that AnxA2 inhibition or depletion, using antibodies or knockdown approaches, strongly reduced insulin-inducible GLUT4 translocation [51]. As insulin exposure promoted GLUT4, but not AnxA2, trafficking to the cell surface [55], it appears unlikely that direct interaction or GLUT4 translocation along AnxA2-positive vesicles occurs. Alternatively, the underlying mechanism could involve a possible role of AnxA2 in insulin signaling through the modulation of insulin receptor internalization [46]. Indeed, the fact that insulin induced AnxA2 phosphorylation [46,56,60], AnxA2 sumoylation [50], and enhanced AnxA2 secretion [60] further indicates that the expression, localization and activity of AnxA2 is closely connected to insulin signaling and glucose handling in adipocytes. Hence, further studies are needed to clarify these current gaps of knowledge and discrepancies.

Secondly, AnxA2 has also been associated with fatty acid accumulation. In fact, in endothelial cells and adipocytes of white adipose tissue, AnxA2 was critical for the cellular uptake of fatty acids. AnxA2 was found to bind prohibitin and the fatty acid transporter CD36 in both cell types, and assembly of this complex at the plasma membrane was enforced by the presence of fatty acids [57]. This protein complex not only improved fatty acid uptake in these two often neighbouring cell types, but also enabled the transport of fatty acids from the endothelium to adipocytes. In further support of these observations, palmitate-inducible expression of inflammatory genes like IL-6, IL-1 beta and tumor necrosis factor alpha was markedly diminished upon AnxA2 suppression, while AnxA2 overexpression amplified the proinflammatory capacity of this saturated fatty acid [59].

Several in vivo studies addressed the aforementioned potential roles of AnxA2 in glucose and fatty acid metabolism (Figure 2). However, AnxA2 null mice had reduced steady-state glucose levels and a normal glucose tolerance [57]. As the glucose uptake of white adipose tissues was comparable in the control and AnxA2-deficient animals, it was concluded that AnxA2 did not have a central function in GLUT4 translocation in vivo. On the other hand, AnxA2-deficient animals had a delayed clearance of infused fatty acids, indicating that the lack of AnxA2 compromised CD36-mediated removal of fatty acids from the bloodstream [57]. Given that thermogenic activation of brown adipose tissue accelerated CD36-dependent clearance of plasma triglycerides [13], and palmitoylation-dependent CD36 localization and trafficking in adipose tissue being sensitive to acute cold exposure [113], testing cold tolerance in AnxA2 KO-mice in future studies could provide further critical insight. Taken together, these findings might point at AnxA2 contributing to a more rapid clearance of lipids and the improvement of postprandial hyperlipidemia.

More recently, others investigated adenoviral-mediated AnxA2 up- or downregulation in mice fed a high fat diet. Animals with low AnxA2 levels had reduced body weight at the end of the study, displaying improved fasting blood glucose and insulin levels, as well as glucose and insulin tolerance. Overexpression of AnxA2 did not change any of these parameters. In addition, AnxA2 depletion was associated with less adipose tissue macrophages and inflammation, which was enhanced by AnxA2 overexpression [59]. Hence, several AnxA2 functions observed in cell-based studies might be relevant in stress-induced conditions in vivo, and, as outlined above, cell and animal studies support an involvement of AnxA2 in adipose tissue function. In line with these observations, expression studies in humans and animal models suggest that AnxA2 levels are tightly regulated, often responding to changes in whole body and adipose tissue physiology. For instance, AnxA2 was detected in preadipocytes and was modestly induced during 3T3-L1 adipogenesis [54]. Guggulsterone, a natural drug which inhibits adipocyte differentiation and induces apoptosis, increased AnxA2 expression in these cells [54]. While AnxA2 mRNA remained unchanged, post-translational processing of AnxA2 protein was induced by guggulsterone [54], indicating that truncated AnxA2 isoforms may exert so far unknown inhibitory functions during adipogenesis. However, in another study, troglitazone-induced PPARγ activation, which promotes adipocyte differentiation [99], upregulated AnxA2 mRNA and protein expression in 3T3-L1 adipocytes [51]. Likewise, the PPARγ agonist rosiglitazone also induced AnxA2 levels in subcutaneous fat of obese but otherwise healthy men. Together with the abovementioned studies suggesting AnxA2 to promote glucose and fatty acid uptake, one can speculate that this drug-induced upregulation of AnxA2 may contribute to the beneficial therapeutic effects of the rosiglitazone-induced lowering of fasting insulin, glucose, and free fatty acids in plasma [33].

Interestingly, in murine adipose tissue AnxA2 protein levels were approximately two-fold higher in large compared to small adipocytes. This differential expression pattern was abrogated in fat-specific insulin receptor knock-out mice [47]. As the increased size of adipocytes is associated with an elevated capacity for insulin-inducible neutral lipid storage, this further supports a function of AnxA2 upregulation in insulin-dependent metabolic changes during adipocyte differentiation and growth. Indeed, a comparison of wild type and AnxA2-deficient mice revealed that AnxA2 was essential for adipocyte growth, whereas adipogenesis was unaffected by the loss of AnxA2 [57].

Proteomic approaches to identify changes in weight loss and physical activity identified altered AnxA2 levels in adipose tissue. Although a two-week high-intensity intermittent training of overweight men neither improved BMI nor the parameters of insulin sensitivity, the inflammatory marker IL-6, as well as AnxA2 and fatty acid synthase were significantly reduced in subcutaneous fat [53]. Likewise, dietary changes, such as a low-fat, high-complex carbohydrate diet supplemented with long-chain n-3 polyunsaturated fatty acids not only improved glucose and fatty acid metabolism, but also downregulated AnxA2 expression in subcutaneous fat [52]. In contrast, a five week very low calorie diet improved metabolic health and BMI of obese subjects, yet AnxA2 and GLUT4 levels increased, whereas CD36 expression declined, in subcutaneous adipose tissues [49]. AnxA2 was also higher in subcutaneous fat after weight loss achieved by a very low calorie diet [48]. In summary, the human studies listed here do not consistently imply a common theme that associates similar changes of AnxA2 levels in fat tissues upon weight loss. Likewise, discordant findings were also published on the regulation of AnxA2 expression in murine obesity. Here, AnxA2 was expressed in epididymal and mesenteric fat. Diet-induced obesity led to elevated AnxA2 protein levels in both fat depots, which was also increased in the liver and skeletal muscle [59]. In spite of this, hepatic AnxA2 protein amounts were found to be reduced in mice fed a high fat diet in a separate study [58]. Thus, for a clearer picture of potential AnxA2 functions in fat tissue (Table 1), more studies are needed to improve our understanding of the regulation of AnxA2 protein expression, localization and interaction partners in adipocytes and other cells of fat tissues.

### 2.3. Annexin A6 (AnxA6)

AnxA6 is found in most cells and tissues, with abundant levels being expressed in endothelial and endocrine cells, hepatocytes and macrophages [23,25,26,27,114]. The plasma membrane and endocytic compartments represent the most common AnxA6 localizations [26,30,63,66,115,116,117,118], but AnxA6 is also found along the secretory pathway [23,25,119], mitochondria [120] and lipid droplets [61,121]. Like other annexins, and depending on the cellular localization and repertoire of interaction partners, AnxA6 participates in many cellular activities, some of which potentially relevant for adipose tissue function, such as endo- and exocytosis, signal transduction, cholesterol homeostasis, stress response [23,25,26,27,30,64,94,116] and lately, neutral lipid accumulation [61,62].

The multifunctionality of AnxA6 has made it difficult to assign specific AnxA6 functions to particular cell types, but despite a still limited number of studies addressing AnxA6 in adipocyte biology, several cellular processes that are modulated by AnxA6 could possibly be relevant for adipocyte function (Table 1). To begin with, AnxA6 upregulation inhibits cholesterol export from late endosomes, which perturbs cellular cholesterol homeostasis similar to mutations in the late endosomal/lysosomal NPC1 cholesterol transporter. This leads to reduced cholesterol levels in other compartments, such as the plasma membrane, Golgi apparatus and recycling endosomes [63,73]. Consequently, membrane trafficking is compromised, and we initially observed reduced numbers of caveolae due to caveolin-1 accumulation in the Golgi [63]. This could be highly relevant for adipocyte function, as caveolae are most prominent in adipocytes, with roles in endocytosis, cholesterol and fatty acid uptake, lipid droplet formation and signal transduction [122,123]. In follow-up studies, we then identified AnxA6-induced cholesterol imbalance to cause mislocalization and dysfunction of several cholesterol-sensitive SNARE proteins in the secretory pathway [72,73], all of which are fundamental for the metabolic response that facilitates GLUT4 translocation in adipocytes [124]. Also, recent findings from our laboratories indicate that cholesterol accumulation in late endosomes of NPC1 mutants promotes the interaction of AnxA6 with the Rab7-GTPase activating protein TBC1D15 (Rentero, Grewal and Enrich, unpublished results), which has recently been implicated in Rab7-dependent pathways that regulate GLUT4 translocation to the cell surface [125]. Impaired insulin signaling and glucose uptake in 3T3-L1 adipocytes upon NPC1 inhibition [19] extend support for a model of upregulated AnxA6, through cellular cholesterol imbalance, to impact on GLUT4 trafficking.

Secondly, AnxA6 associates with secretory granules in a Ca^2+^-dependent manner [67], participates in Ca^2+^ homeostasis through store-operated Ca^2+^ entry [70] and alters catecholamine secretion [71], all of which with links to the secretory pathway that enables adiponectin release [126]. Thirdly, the scaffolding function of AnxA6 is critical for the formation and activity of several signalling complexes [26,29,64,65,66], with roles in GLUT4 translocation and lipolysis [127]. Finally, we recently identified the association of AnxA6 with lipid droplets in hepatocytes to influence their capacity to store neutral lipids [61], which could also be relevant for neutral lipid storage in fat tissue.

Initial insights into AnxA6 functions in fat tissue were lately obtained from the characterization of differentiated 3T3-L1 adipocytes overexpressing or lacking AnxA6. In this model, siRNA-mediated AnxA6 knockdown impaired preadipocyte proliferation. Moreover, maturation of AnxA6-depleted 3T3-L1 adipocytes was associated with increased storage of triglycerides and elevated release of adiponectin [68]. The latter finding was not observed in oleate-loaded cells [68], possibly indicating independent mechanisms that cause changes in triglyceride accumulation and adiponectin release upon AnxA6 depletion. Vice versa, AnxA6 overexpression in 3T3-L1 cells lowered cellular triglycerides and adiponectin release (Figure 2). In addition, the catecholamine-stimulated phosphorylation of hormone-sensitive lipase (HSL) to promote lipolysis was impaired in AnxA6-depleted cells and coincided with AnxA6 localization on lipid droplets in adipocytes, implicating a scaffolding function of AnxA6 at the lipid droplet membrane possibly relevant for HSL phosphorylation and not directly linked to adiponectin release through the secretory pathway [68]. Importantly, this function of HSL is not critical for fatty acid metabolism in non-adipose tissue [128], which might contribute to explain the opposite effects of AnxA6 up- or downregulation on neutral lipid storage in cells from liver and fat tissue [61,68]. Notably, AnxA6 levels did not change lipopolysaccharide response of 3T3-L1 adipocytes, and basal as well as lipopolysaccharide-induced IL-6 levels were comparable in groups with high and low AnxA6 levels [68].

Follow-up studies in AnxA6 KO-mice, which have normal body weight, glucose and insulin levels, support some of the cell-based studies summarized above. In particular, serum adiponectin levels were higher, while reduced amounts of adiponectin were found in the subcutaneous fat of the AnxA6 KO-animals [68]. As cholesterol is critical for the release of adiponectin through the secretory pathway [20], we speculate that the regulatory role of AnxA6 in cholesterol homeostasis [31,63,64,72,73] could be responsible for alterations in adiponectin plasma levels in the AnxA6 KO-animals.

Despite increased lipid storage in AnxA6-depeleted 3T3-L1 adipocytes, circulating triglycerides, free fatty acids and cholesterol were normal in AnxA6 KO-mice [68]. Systemic lipid levels of AnxA6-deficient animals were also comparable to controls after a high fat diet for 17 weeks [61,62]. Most interestingly, AnxA6 KO-mice gained less adipose tissue during high fat feeding [61,62], which might be in line with the impaired proliferation observed in AnxA6-depleted preadipocytes [68]. On the other hand, and in contrast to the cell-based studies described above [68], circulating leptin and adiponectin levels were slightly reduced in high fat diet fed AnxA6 KO-mice [61,62]. Lower fat mass is usually associated with higher adiponectin and improved glucose homeostasis [21]. Such improvements were not observed in the AnxA6 KO-animals [61,62] and AnxA6-related functions in other organs may need to be considered to possibly explain these up till now opposing obervations.

Strikingly, AnxA6 deficiency in mice compromised regulatory steps to downregulate hepatic gluconeogenesis that only became apparent after high fat diet feeding [61,62]. Likewise, dysfunctional hepatic glucose homeostasis in AnxA6-KO mice was also observed after induction of metabolic stress upon partial liver hepatectomy or starvation [129]. Given the prominent role for adipokines in the coordination of hepatic glucose homeostasis, we speculate that so far unidentified changes in the inter-organ metabolic communication beween fat and liver tissues of AnxA6-KO mice may contribute to the fine-tuning of hepatic glucose metabolism, with potential consequences for the systemic control of glucose in health and disease.

The abovementioned and in-part profound effects of AnxA6 up- or downregulation on the central aspects of adipocyte function, including growth, lipid storage, and adiponectin release, suggest that complex mechanisms might be in place to control AnxA6 expression levels in fat tissue. Several studies provide some insight in this matter. AnxA6 protein expression modestly increased during 3T3-L1 cell adipogenesis and was clearly induced in mature human adipocytes when compared to their respective preadipocytes [68]. AnxA6 protein levels were, however, not changed upon lipid loading of adipocytes [68]. Likewise, exposure to high glucose or lipopolysaccharide did not impact on AnxA6 protein levels in 3T3-L1 cells. On the other hand, oxidative stress, which suppresses adiponectin release and contributes to insulin resistance in obese adipose tissues [130], was associated with AnxA6 upregulation in 3T3-L1 adipocytes [68].

AnxA6 was also highly expressed in human monocytes, which can infiltrate fat tissue, and further increased in monocytic cells of overweight patients [74]. How this might impact adipose tissue function is still unclear, but high AnxA6 levels in phagocytes may accompany the process leading to foam cell formation and atherosclerosis [64,74]. Alternatively, AnxA6-induced changes in membrane order at the plasma membrane [131] may influence the distribution and activity of lipoprotein receptors and cholesterol transporters at the cell surface responsible for cholesterol efflux [132]. Of note, adiponectin, which protects from cardiovascular diseases [74,133], reduced AnxA6 protein expression in human monocytic cells [74,133], but not in 3T3-L1 adipocytes [68].

Complex and differential AnxA6 expression patterns have also been observed in animal and human studies. In subcutaneous, perirenal and epididymal adipose tissues from mice fed a high fat diet for 14 weeks, AnxA6 levels were strongly induced [68]. This may in part be related to increased AnxA6 expression in macrophages [74]. Additionally, obese murine adipocytes also displayed higher AnxA6 protein levels [68]. AnxA6 protein amounts remained unchanged in the visceral fat of overweight patients when compared to normal weight patients, illustrating that AnxA6 levels do not increase when body weight and adipocyte size grow in humans [68]. Furthermore, AnxA6 expression was induced in adipocytes during aging [35], which is associated with oxidative stress and a decline in adipocyte function [134]. Hence, increased reactive oxygen species rather than cell hypertrophy seem to mediate the upregulation of AnxA6 in adiposity.

Visceral fat accumulation has deleterious effects [2,68,69] and AnxA6 protein levels were higher in human and murine visceral compared to subcutaneous adipose tissues [2,68,69]. In adipocytes purified from the respective human fat depots, AnxA6 protein amounts were also more abundant in the visceral fat cells [68]. Remarkably, fat depot distribution of AnxA6 changed in obesity. Here, AnxA6 levels were higher in subcutaneous adipose tissues compared to intraabdominal fat. Such a change in fat depot expression is uncommon and cell-type specific regulation of AnxA6 may need to be evaluated to identify the underlying mechanisms.

In murine epididymal fat, AnxA6 protein levels were approximately 60% higher in large when compared to small adipocytes. This size-dependent change in AnxA6 expression was only detected in insulin receptor knock-out mice but not in the respective control animals [47]. This indicates that adipocyte growth is not associated with higher AnxA6 expression as long as the cells respond to insulin [47]. Whether a differential insulin response of subcutaneous and visceral adipocytes [2] contributes to altered AnxA6 protein needs further studies. Insulin did not change AnxA6 protein levels in 3T3-L1 cells, excluding a direct effect of this hormone [68].

Finally, in brown fat, which is quite distinct from other fat tissues as its main function is to produce heat, AnxA6 protein amounts remained unchanged in obesity [68]. Altogether, AnxA6 is differentially expressed in the various fat depots and in some cases, response to diet was observed (Table 1). Altogether this may indicate differential AnxA6 functions in the various fat tissues, which still need to be resolved in future studies.

### 2.4. Other Annexins

#### 2.4.1. Annexin A3 (AnxA3)

In comparison to the depth of literature on AnxA1, A2 and A6, up to date only a limited number of studies have examined AnxA3 expression and function (Table 1). AnxA3 is most prominent in neutrophils and macrophages and was detected in heart, lung, placenta, kidney and spleen, with highest levels in murine adipose tissue [77]. Besides its Ca^2+^-dependent membrane binding behavior, its intra- and extracellular locations and physiological functions are still poorly understood. Most AnxA3-related studies focussed on its potential as a biomarker in several cancers and the association of AnxA3 with chemotherapy resistance [135], with possible roles in the proliferative and invasive properties of cells. Interestingly, a recent study identified the recruitment of AnxA3 to lipid droplets of hepatitis C virus infected Huh7 hepatocytes [76], facilitating the interaction of viral proteins with apolipoprotein E (ApoE) during virus maturation and egress. Given the prominent role of ApoE in mouse and human adipocyte differentiation and lipid accumulation [136], one can speculate that yet to be identified environmental signals may also trigger AnxA3-driven interactions with ApoE or other proteins on the lipid droplet membrane during fat cell differentiation. In fact, one report identified AnxA3 to negatively regulate adipogenesis. In this study, AnxA3 protein was highly expressed in preadipocytes and strongly downregulated during 3T3-L1 cell differentiation [77]. Marked suppression of AnxA3 in early adipogenesis suggested an inhibitory function of AnxA3 in adipocyte differentiation [77]. Indeed, when AnxA3 was depleted by siRNA transfection of preadipocytes, expression of PPARγ2 and lipid droplet accumulation were increased, enhancing terminal adipocyte differentiation [77]. Of note, AnxA3 mRNA levels were comparable in the different white fat depots [77] (Table 2), indicating similar roles in the various fat locations. Interestingly, stromal vascular cell fractions expressed higher AnxA3 mRNA levels compared to adipocytes [75]. Analysis of publicly accessible DNA microarray data confirmed higher AnxA3 mRNA expression in murine stromal vascular cells (Table 2), suggesting that AnxA3 might fulfill multiple cell-specific functions in fat tissue.

#### 2.4.2. Annexin A5 (AnxA5)

AnxA5 is the most abundant annexin and except in neurons, is expressed ubiquitously [25,137]. During proliferation, differentiation and in many cancers, AnxA5 levels are often up- or downregulated [138]. Ca^2+^ elevation triggers AnxA5 binding to various cellular sites [23,139,140] to participate in cell growth and death, Ca^2^ signalling and homeostasis, membrane domain organization and transport [23,30,140]. Therapeutically relevant, extracellular AnxA5 binds to outer membrane phosphatidylserine, allowing detection of apoptotic cells [137]. Furthermore, AnxA5 has prominent extracellular roles in blood coagulation, phagocytosis, viral infection, membrane invagination and membrane repair [137,138,141,142].

Although one can envisage several of these intra- and extracellular functions listed above being relevant for the proper functioning of several cell types in adipose tissue, such as macrophages, endothelial and vascular smooth muscle cells as well as adipocytes, current knowldege on AnxA5 function in fat physiology is still insignificant (Table 1). Analysis of publicly available expression data (Geo profiles) revealed that AnxA5 mRNA was mostly expressed in murine adipocytes when compared to the stromal vascular cells in subcutaneous and intraabdominal fat. Whereas adipocytes of both fat depots had similar AnxA5 mRNA levels, stromal vascular cells in subcutaneous fat expressed less AnxA5 mRNA (Table 2). In line with other differentiation models, AnxA5 protein expression was induced in adipocytes during aging [35] and one study demonstrated an association of AnxA5 polymorphisms with obesity in a Korean patient cohort [78], which may suggest a function of AnxA5 directly or indirectly contributing to fat deposition, storage or mobilization.

#### 2.4.3. Annexin A7 (AnxA7)

AnxA7 is the only annexin that contains a long (100 amino acids) and hydrophobic N-terminus. Due to alternative splicing, a 47 kD splice variant is found in most tissues, while a larger 51 kDa isoform is expressed in the brain, heart and skeletal muscle [143]. In these various cells and organs, Ca^2+^-inducible association of AnxA7 with secretory vesicles, the plasma membrane and the nuclear envelope has been observed [144], with possible roles in Ca^2+^/GTP-dependent exocytic pathways, prostaglandin synthesis, cardiac remodelling and inflammatory myopathies [81,145,146]. In addition, the GTPase activity of AnxA7 has potential as a tumour suppressor in several cancers [147].

The AnxA7 functions listed above and related to membrane transport, Ca^2+^ signalling and hormone production could be relevant in fat, but very little is still known about potential roles for AnxA7 in adipocytes or other cell types in this tissue (Table 1). Nevertheless, in other cells and tissues, several AnxA7-related tasks may also influence adipose tissue function. For example, one mouse model lacking AnxA7 displayed defects in Ca^2+^ release and Ca^2+^-dependent signal transduction, affecting insulin secretion [82,83]. On the other hand, another independently generated AnxA7 KO-mouse model was strikingly different and did not reveal a role for AnxA7 in Ca^2+^-dependent insulin secretion [79]. In addition, in some cell types, AnxA7 negatively regulates cyclooxygenase-dependent prostaglandin E2 formation [80]. Hence, elevated plasma prostaglandin levels in AnxA7 KO-mice may contribute to decreased glucose tolerance and elevated glucose-inducible insulin secretion [81]. Most relevant for fat tissue in obesity, cyclooxygenase-dependent prostaglandin E2 production has been associated with pathologic complications that lead to inflammation and fibrosis, impaired adaptive thermogenesis and lipolysis in obese white adipose tissue [148].

Although the latter might indicate that AnxA7 plays a role in infiltrating immune cells in dysfunctional adipose tissue, at present, very limited information on AnxA7 expression patterns in normal and obese fat tissue is available. AnxA7 mRNA was comparable in adipocytes and stromal vascular cells in subcutaneous and intraabdominal fat (Table 2). Evidently, more studies are needed to possibly identify yet unknown AnxA7 functions in fat tissue.

#### 2.4.4. Annexin A8 (AnxA8)

AnxA8 was first identified in human placenta [149] and, with the exception of acute promyelocyte leukemia [150], is only expressed at low levels in lung, skin, liver, and kidney [151]. Earlier reports described AnxA8 to inhibit blood coagulation [152], but cellular AnxA8 localizations and functions are still not fully understood. AnxA8 may provide opportunities as a biomarker in several cancers [153,154,155] and more recently, has been linked to the transdifferentiation of retinal pigment epithelial cells [156].

Nonetheless, within the context of adipose tissue function, the unique affinity of AnxA8 towards phosphatidylinositides and F-actin relevant for membrane-cytoskeleton interactions, might be most important. In fact, these distinctive membrane- and actin-binding properties of AnxA8 affect the functioning of late endosomes [84,85] and in endothelial cells, this contributes to control the delivery of CD63 from late endocytic vesicles to the cell surface for leukocyte recruitment and migration [86,89]. In addition, AnxA8 is associated with cholesterol-rich late endosomes, and similarly to AnxA6 overexpression or NPC1 inhibition [19,31,63,64,72,73], AnxA8 depletion results in cholesterol accumulation in this compartment [87]. This may indicate a coordinated mechanism to control AnxA8 and AnxA6 expression levels and their relative amounts in the late endosomal compartment. Along these lines, and as discussed for AnxA6 overexpression and NPC1 deficiency above (see 2.3.), it is tempting to speculate that late endosomal cholesterol accumulation triggered by AnxA8 downregulation might compromise adipocyte function leading to the improper performance of molecular events in caveolae [63,122,123], or related to insulin signaling and GLUT4 translocation [19,72,73,124].

The findings described above suggest that changes in AnxA8 expression levels may cause cellular dysfunction However, little is so far known if AnxA8 expression levels correlate with metabolic complications in obese fat tissue (Table 1). AnxA8 mRNA was expressed in human adipose tissues and was similar in subcutaneous and visceral fat depots of obese men [88]. In mice, AnxA8 mRNA levels were higher in adipocytes than stromal vascular cells. Interestingly, AnxA8 expression in adipocytes was more abundant in subcutanoues fat depots (Table 2). This observation adds AnxA8 to the list of candidate proteins that are differentially expressed in the various fat depots, and possibly relevant to further evaluate the deleterious effects of visceral adiposity [88].

#### 2.4.5. Other Annexins

Up to date, it is unknown if the remaining annexins AnxA4, A9, A10, A11 and A13 contribute to the proper functioning of adipose tissue (Table 1). Out of those annexins, current literature has associated AnxA4 with cAMP production, which could be relevant for lipolysis [90]. Also, roles for AnxA11 in exocytosis and cytokinesis could influence fatty acid release or adipokine secretion [91]. Of note, expression data for AnxA4, AnxA9 and AnxA13 in fat was not publicly available, but AnxA10 and A11 are indeed expressed in fat tissue. AnxA10 expression was similar in adipocytes and stromal vascular cells in subcutaneous and intraabdominal fat (Table 2). AnxA11 was mostly expressed in the stromal vascular cells of subcutaneous adipose tissues when compared to the respective adipocytes and to intraabdominal stromal vascular cells (Table 2). Differential levels of AnxA11 mRNA between the cell populations did not exist in intraabdominal adipose tissue (Table 2). Further studies are evidently required to unravel their possible functions in fat tissue.

## 3. Conclusions

Annexins bind negatively charged phospholipids and cholesterol in a Ca^2+^-dependent and reversible manner, and together with transient interactions with membrane-associated proteins, this contributes to dynamic changes in the structural and functional organization of membrane domains [23,25,26,27,28,29,30,31,64,93,94,109,116,118,137,157]. As outlined in this review, this membrane organizing function of annexins also seems highly relevant for adipose tissue physiology. In particular, AnxA2 has been linked to the insulin-dependent translocation of GLUT4 as well as CD36-mediated fatty acid uptake [51,57,59], the latter providing a protective function in the postprandial state. Likewise, AnxA6 affects signaling events relevant for lipid storage and, importantly, regulates adiponectin release, an essential adipokine in metabolic health [61,62,64,68,70]. Alternatively, the most prominent disease-preventing functions of AnxA1 in adiposity, glucose and lipid homeostasis are facilitated through its extracellular activity as a FPR2 ligand [34,37,39,41,42,45,104]. While functions and mechanistic insights for these three annexins in fat tissue are emerging, up until now all other annexins have been barely studied in the context of obesity, adipocyte physiology, and adipokine production. Future experiments, combining biochemical and imaging techniques in overexpression and knockdown cells and animal models, together with high-throughput and innovative technologies addressing transcriptomics, proteomics, lipidomics, and metabolomics in adipocyte-specific knock-out models may identify the impact of individual annexins in the molecular pathways that contribute to dysregulated adipokine production and fat cell function in obesity.

## Figures and Tables

**Figure 1 ijms-20-03449-f001:**
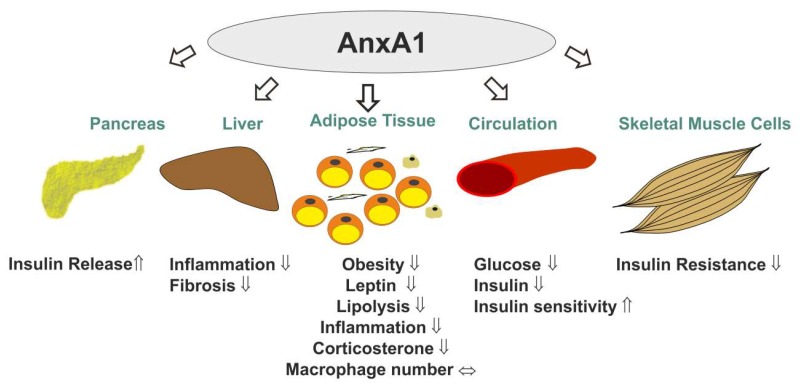
The multiple roles of AnxA1 in metabolism. AnxA1 increased insulin release of pancreatic beta-cells [108] and improved insulin response of skeletal muscle and whole body insulin sensitivity, thus lowering circulating glucose and insulin levels [34,45]. AnxA1 further ameliorated hepatic inflammation and fibrosis in a murine NASH model [104]. AnxA1 null mice were more obese, produced more leptin and had higher adipose tissue lipolysis, inflammation and corticosterone levels. AnxA1 did not alter the recruitment of adipose tissue macrophages [34].

**Figure 2 ijms-20-03449-f002:**
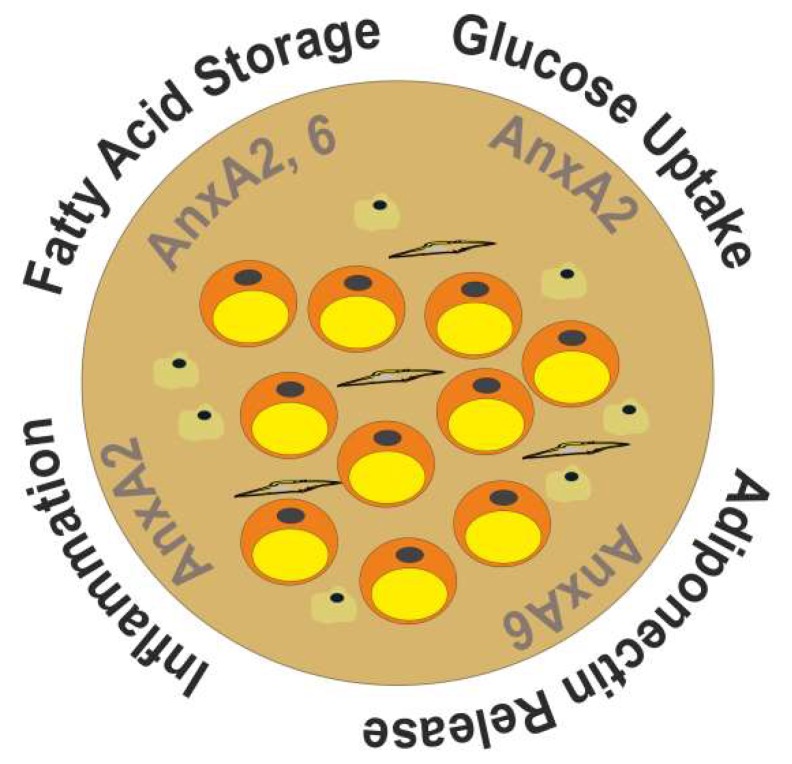
The diverse roles of AnxA2 and AnxA6 in adipose tissue function. AnxA2 improves uptake and storage of fatty acids [57] and may have a role in glucose uptake and adipose tissue inflammation [51,59]. On the other hand, AnxA6 modulates lipolysis and adiponectin secretion [68].

**Table 1 ijms-20-03449-t001:** Domain structure, expression patterns, and potential functions of annexins expressed in adipose tissue. The different length of the N-terminal leader and C-terminal annexin repeats 1–4 (1–8 for AnxA6) for each annexin are indicated. AnxA13a differs from AnxA13b by a 41 amino acid N-terminal deletion [32]. Relevant references for each annexin are listed. AnxA, annexin; GLUT4, glucose transporter type 4; HFD, high-fat diet; HSL, hormone-sensitive lipase; SV, stromal-vascular fraction; TZDs, thiazolidinediones. N/A, not available.

Name	Structure	Adipose Tissue Expression	Function	References
**A. Prominent Annexins in Adipose Tissue.**
AnxA1	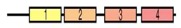	adipocytes, SV, visceral fat, subcutaneous fat, obesity ↑,HFD ↑, TZDs ↑	insulin response ↑, obesity ↓,leptin ↓,inflammation ↓	[33,34,35,36,37,38,39,40,41,42,43,44,45]
AnxA2	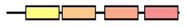	adipocytes, endothelial cells, macrophages, subcutaneous fat, epididymal fat, mesenteric fat, guggulsterone ↑, TZDs ↑	GLUT4 translocation, insulin response, glucose uptake, CD36-mediated fatty acid uptake, inflammation ↑, macrophage infiltration ↑, HSL activation	[46,47,48,49,50,51,52,53,54,55,56,57,58,59,60]
AnxA6	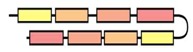	adipocytes, macrophages, subcutaneous fat, perirenal fat, epididymal fat, visceral fat, brown fat,obesity ↑, HFD ↑, oxidative stress ↑	preadipocyte proliferation ↑, triglyceride storage ↓, adiponectin release ↓, cholesterol-dependent caveolae formation, cholesterol-dependent GLUT4 translocation, cholesterol-dependent adiponectin secretion?	[2,35,47,61,62,63,64,65,66,67,68,69,70,71,72,73,74]
**B. Other Annexins in Adipose Tissue.**
AnxA3	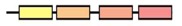	adipocytes, SV, subcutaneous fat, intraabdominal fat	adipocyte differentiation ↓,lipid accumulation?	[75,76,77], Geo Profiles; DataSet Record GDS2818
AnxA5	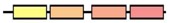	SV, subcutaneous fat, intraabdominal fat	fat deposition, storage or mobilization?	[35,78],Geo Profiles; DataSet Record GDS2818
AnxA7	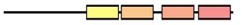	SV, subcutaneous fat, intraabdominal fat	infiltration of immune cells in dysfunctional adipose tissue?	[79,80,81,82,83], Geo Profiles; DataSet Record GDS2818
AnxA8	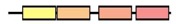	adipocytes, SV, subcutaneous fat, intraabdominal fat	cholesterol-dependent caveolae formation, cholesterol-dependent GLUT4 translocation, cholesterol-dependent adiponectin secretion?	[84,85,86,87,88,89], Geo Profiles; DataSet Record GDS2818
**C. Insufficiently Studied Annexins in Adipose Tissue.**
AnxA4	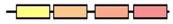	N/A	lipolysis?	[90]
AnxA9	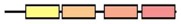	N/A	?	
AnxA10	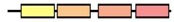	adipocytes, SV, subcutanous fat, intraabdominal fat	?	
AnxA11	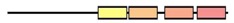	adipocytes, SV, subcutanous fat, intraabdominal fat	fatty acid release, adipokine secretion?	[91]
AnxA13a	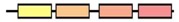	N/A	?	
AnxA13b	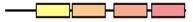	N/A	?	

**Table 2 ijms-20-03449-t002:** Expression of annexins AnxA3, A5, A7, A8, A10 and A11 mRNA in murine adipose tissues. Analysis of publicly accessible DNA microarray data (Geo Profiles; DataSet Record GDS2818) was done with unpaired Students t-test. A p-value < 0.05 was regarded as significant. 
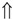
, 
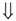
 and 
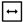
 indicate higher, lower and unchanged mRNA levels, respectively, in adipocytes relative to stromal vascular cells (SVC) or in subcutaneous (sc) fat compared to intraabdominal (intra) fat. The mRNA expression data for AnxA4, AnxA9 and AnxA13 in fat tissue were not available.

	Subcutaneous FatAdipocyte/SVC	Intrabdominal FatAdipocyte/SVC	AdipocytesSc/Intra	SVCSc/Intra
AnxA3	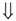	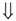	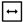	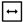
AnxA5	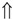	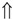	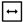	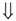
AnxA7	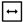	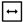	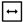	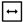
AnxA8	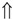	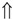	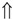	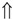
AnxA10	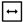	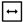	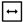	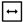
AnxA11	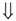	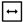	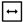	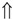

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
