# Peer review of "Annexins in Adipose Tissue: Novel Players in Obesity"

_ijms, 2019, doi:10.3390/ijms20143449_

Reviewer 1 Report

This is an interesting review about the role of annexins in obesity and co-morbidities mainly focused on adipose tissue biology. I have some suggestions and comments for the authors:

1) Page 2, line 52: When they talk about stromovascular fraction, they just mention macrophages as immune cells, but what about T cells, B cells etc. Any data about that?

2) Page 2, line 60: "...recent identification of brown fat in humans...". It is not really recent, 2009. The authors should cite the 3 main papers regarding that issue.

In this same paragraph, the authors should talk about the endocrine role of brown adipose tissue (Villarroya et al., 2013 etc),

3) Comments about review sections:

The authors should make a big table 1 with all the annexins to describe briefly the structure, cells where are expressed and main functions (with citations).

I suggest to the authors to change sections. The introduction about obesity is fine, but when they talk about each annexin, the review is hard to follow, and same structures are repeted. So I think it would be easier to read if the sections were something like: a) adipocyte differentiation b) glucose homeostasis, c) lipid metabolism d) Inflammation e) liver disease. Then, the authors can include main annexins 1, 2 and 6 together in each section. After that, they can talk about other annexins 3, 5, 7, 8 ...

Author Response

This is an interesting review about the role of annexins in obesity and co-morbidities mainly focused on adipose tissue biology. I have some suggestions and comments for the authors:

1) Page 2, line 52: When they talk about stromovascular fraction, they just mention macrophages as immune cells, but what about T cells, B cells etc. Any data about that?

We agree with the reviewer that this is an interesting point and to include the contribution of immune cells in adipose tissue biology during obesity, we have added the following to this section (Page 2, lane 63-65) as follows:

‘In addition, in adipose tissue other myeloid cells, as well as T- and B-lymphocytes, have been linked to macrophage homeostasis and the inflammatory process associated with obesity (Ivanov S, Merlin J, Lee MKS, Murphy AJ, Guinamard RR. Atherosclerosis. 2018 Apr;271:102-110).‘

The new reference was added to the reference list and the numbering of references was changed accordingly.

2) Page 2, line 60: "...recent identification of brown fat in humans...". It is not really recent, 2009. The authors should cite the 3 main papers regarding that issue.

As pointed out by this reviewer, brown fat was identified more than 10 years ago. The text was changed accordingly and the three main references related to this finding were added  to this section (Page 2, Lane 73-75) as follows:

‘Additionally, the identification of brown fat in humans (Saito et al., Diabetes 58:1526–1531, 2009; van Marken Lichtenbelt et al., N Engl J Med 360:1500–1508, 2009; Virtanen et al., N Engl J Med 360:1518–1525, 2009) has initiated new exciting research in the field in the last decade, as…‘

The new references were added to the reference list and the numbering of references was changed accordingly.

In this same paragraph, the authors should talk about the endocrine role of brown adipose tissue (Villarroya et al., 2013 etc)

As requested by this reviewer, a sentence on the endocrine role of brown tissue, including three new references was added  to this section (Page 2, Lane 84-88) as follows:

‘Moreover, especially under thermogenic stimulation, brown fat releases several bioactive factors with endocrine properties, including insulin-like growth factor I, interleukin-6, or fibroblast growth factor-21, which influence hepatic and cardiac function, contributing to improved glucose tolerance and insulin sensitivity (Villarroya et al., Am J Physiol Endocrinol Metab. 2013 Sep 1;305(5):E567-72: Villarroya et al., Nat Rev Endocrinol. 2017 Jan;13(1):26-35; Scheja L, Heeren J. J Hepatol. 2016 May;64(5):1176-1186).‘

The new references were added to the reference list and the numbering of references was changed accordingly.

3) Comments about review sections:

The authors should make a big table 1 with all the annexins to describe briefly the structure, cells where are expressed and main functions (with citations).

We absolutely agree with this reviewer that an additional table summarizing the key features of the annexins discussed in this review would be helpful. To improve the overall understanding and reading experience, we now provide the new Table 1 showing the structure of those annexins relevant for this review, their expression patterns, potential functions and relevant references.

I suggest to the authors to change sections. The introduction about obesity is fine, but when they talk about each annexin, the review is hard to follow, and same structures are repeted. So I think it would be easier to read if the sections were something like: a) adipocyte differentiation b) glucose homeostasis, c) lipid metabolism d) Inflammation e) liver disease. Then, the authors can include main annexins 1, 2 and 6 together in each section. After that, they can talk about other annexins 3, 5, 7, 8 ... 

As outlined below, we believe that the separate chapters on AnxA1, A2 and A6 provide independent sections that each deliver a comprehensive overview of a single annexin, addressing the various aspects relevant to fat tissue and obesity. Although this requires an introductory paragraph for each of those annexins with rather general information on gene and protein structure and expression patterns, followed by in part common themes as listed above by this reviewer, it enables researchers with interest in adipose biology or annexin-related aspects to read stand-alone chapters on a single annexin without possibly being distracted by information on other annexins. In addition, the new Table 1 complements the overall content, providing a good overview of key features (structure, expression, function) of all annexins in adipose biology.

We have successfully adopted this approach, as evidenced by significant citation numbers, to describe the roles of individual annexins in the regulation of EGFR trafficking/signalling (Cell Signal. 2009 Jun;21(6):847-58; 131 cites, google scholar), PKC localization/signalling (Cell Signal. 2014 Jun;26(6):1213-25; 48 cites, google scholar) as well as their in vivo roles (Biol Chem. 2016 Oct 1;397(10):1031-53; 27 cites, google scholar). Likewise, other leading researchers in the annexin field have used the separate listing of annexins in their review articles (Am J Physiol Renal Physiol. 2005 Nov;289(5):F949-56; 47 cites, google scholar; Front Pharmacol. 2018 Nov 15;9:1282).

Importantly, Reviewer #2 also considers this review article as well-written and well-structured (see below).

Reviewer 2 Report

In this timely and comprehensive review by Greval et al., the multifaceted roles of annexins in adipocyte biology and obesity development are discussed. The authors begin with a general overview of mechanisms behind obesity-associated comorbidities (mainly insulin resistance) before delving into the specific roles for each of the annexins in this context with reference to cell-based, murine and human data. The review is well-written, well-structured and well-referenced and the figures are clear. I only have several minor suggestions:

Lines 51-68, it may be worth adding that sheer physical cell stress contributes to inflammatory processes and insulin resistance in swelling adipocytes (PMID 23021218). 

Lines 147-153, as the authors state, the resistance to obesity in mice lacking FPR2 is somewhat surprising. It may be worth emphasising in this section that this may be due other ligands of this GPCR apart from AnxA1. 

Lines 233-244, AnxA1 KO were shown to be resistant to obesity in both references 34 and 44. Given these mice had no changes in food intake, it may be worth mentioning the need to perform energy expenditure measurements in these animals in future studies. 

Lines 298-305, the reduction in Cd36 function and fatty acid clearance in Anxa2 KO mice is interesting. Considering that Cd36 is essential for cold tolerance (reference 8 in the manuscript) and regulating it's trafficking through palmitoylation has a similar role (PMID 23021218), it may be worth adding here the importance of testing cold tolerance in Anaxa2 KO mice in future studies. 

Author Response

Comments and Suggestions for Authors

In this timely and comprehensive review by Greval et al., the multifaceted roles of annexins in adipocyte biology and obesity development are discussed. The authors begin with a general overview of mechanisms behind obesity-associated comorbidities (mainly insulin resistance) before delving into the specific roles for each of the annexins in this context with reference to cell-based, murine and human data. The review is well-written, well-structured and well-referenced and the figures are clear. I only have several minor suggestions:

1. Lines 51-68, it may be worth adding that sheer physical cell stress contributes to inflammatory processes and insulin resistance in swelling adipocytes (PMID 23021218). 

This is an interesting point and to include physical stress contributing to inflammation and insulin resistance,  we have added the following on Page 2, lane 59-61, as follows:

‘In fact, even the physical stress triggered by the swelling that occurs in adipocytes upon increased fat accumulation seems to contribute to inflammation and insulin resistance (Ye et al., Cell. 2012 Sep 28;151(1):96-110).‘

The new reference was added to the reference list and the numbering of references was changed accordingly.

2. Lines 147-153, as the authors state, the resistance to obesity in mice lacking FPR2 is somewhat surprising. It may be worth emphasising in this section that this may be due other ligands of this GPCR apart from AnxA1.

As pointed out by this reviewer, to include the potential contribution of other ligands apart from AnxA1, we have added the following on Page 6, lane 170-172, as follows:

‘Although the lack of FPR2 signalling events induced by ligands other than AnxA1 probably also contribute to the phenotype of the FRP2 knock-out mice described above, one can speculate that up- or downregulation of AnxA1 …..‘

3. Lines 233-244, AnxA1 KO were shown to be resistant to obesity in both references 34 and 44. Given these mice had no changes in food intake, it may be worth mentioning the need to perform energy expenditure measurements in these animals in future studies. 

We thank the reviewer for this interesting comment and have included a sentence mentioning these potential future studies (Page 8, lane 272-273) as follows:

‘…, for instance the comparison of energy expenditure measurements in controls and the AnxA1 null mice on chow and high fat diets, is still needed…‘

4. Lines 298-305, the reduction in Cd36 function and fatty acid clearance in Anxa2 KO mice is interesting. Considering that Cd36 is essential for cold tolerance (reference 8 in the manuscript) and regulating it's trafficking through palmitoylation has a similar role (PMID 23021218), it may be worth adding here the importance of testing cold tolerance in Anaxa2 KO mice in future studies. 

We thank the reviewer for this interesting comment and have included a sentence mentioning these potential future studies (Page 9, lane 325-328) as follows:

‘Given that thermogenic activation of brown adipose tissue accelerated CD36-dependent clearance of plasma triglycerides, and palmitoylation-dependent CD36 localization and trafficking in adipose tisse being sensitive to acute cold exposure (Wang et al., Cell Rep. 2019 Jan 2;26(1):209-221.e5), testing cold tolerance in AnxA2 KO-mice in future studies could provide further critical insight. Taken together,…‘

The new reference was added to the reference list and the numbering of references was changed accordingly.